# Shockwave Therapy Combined with Autologous Adipose-Derived Mesenchymal Stem Cells Is Better than with Human Umbilical Cord Wharton’s Jelly-Derived Mesenchymal Stem Cells on Knee Osteoarthritis

**DOI:** 10.3390/ijms21041217

**Published:** 2020-02-12

**Authors:** Chieh-Cheng Hsu, Jai-Hong Cheng, Ching-Jen Wang, Jih-Yang Ko, Shan-Ling Hsu, Tsai-Chin Hsu

**Affiliations:** 1Center for Shockwave Medicine and Tissue Engineering, Kaohsiung Chang Gung Memorial Hospital and Chang Gung University College of Medicine, Kaohsiung 833, Taiwan; t1234@cgmh.org.tw (C.-C.H.); kojy@cgmh.org.tw (J.-Y.K.); shanlin@cgmh.org.tw (S.-L.H.); tsaichin1219@gmail.com (T.-C.H.); 2Department of Orthopedic Surgery, Sports Medicine, Kaohsiung Chang Gung Memorial Hospital and Chang Gung University College of Medicine, Kaohsiung 833, Taiwan; 3Medical Research, Kaohsiung Chang Gung Memorial Hospital and Chang Gung University College of Medicine, Kaohsiung 833, Taiwan; 4School of Nursing, Fooyin University, Kaohsiung 831, Taiwan

**Keywords:** autologous adipose-derived stem cells, Wharton’s jelly-derived mesenchymal stem cells, shockwave therapy, osteoarthritis

## Abstract

Extracorporeal shockwave therapy (ESWT) and mesenchymal stem cells (MSCs) have been reported to have chondroprotective effects in knee osteoarthritis (OA). Here, we examined whether autologous adipose-derived mesenchymal stem cells (ADMSCs) and human umbilical cord Wharton’s jelly-derived mesenchymal stem cells (WJMSCs) increased the efficacy of ESWT in knee OA, and compared the efficacy of the two. The treatment groups exhibited significant improvement of knee OA according to pathological analysis, micro-computed tomography (CT), and immunohistochemistry (IHC) staining. The ADMSCs and ESWT+ADMSCs groups exhibited increased trabecular thickness and bone volume as compared with the ESWT, WJMSCs, and ESWT+WJMSCs groups individually. According to the results of IHC staining, Terminal deoxynucleotidyl transferase dUTP nick end labeling (TUNEL) activity and caspase-3 were significantly reduced in the ADMSCs and ESWT+ADMSCs groups as compared with the WJMSCs and ESWT+WJMSC groups. In mechanistic factor analysis, the synergistic effect of ESWT+ADMSCs was observed as being greater than the efficacies of other treatments in terms of expressions of transforming growth factor (TGF)-β, runt-related transcription factor (RUNX)-2 and sex determining region Y-box (SOX)-9. The type II collagen was expressed at a higher level in the WJMSCs group than in the others. Furthermore, ESWT+ADMSCs reduced the expression of platelet-derived growth factor (PDGF)-BB and increased the expression of bone morphogenetic protein (BMP)-4. Therefore, we demonstrated that ESWT+ADMSCs had a synergistic effect greater than that of ESWT+WJMSCs for the treatment of early knee OA.

## 1. Introduction

Osteoarthritis (OA) is a well-known cartilage disease that involves degradation and loss of articular cartilage [1]. However, OA is usually accompanied by changes in the subchondral bone, with sclerosis, bone cyst, and osteophyte formation [2]. Therefore, the relationships between subchondral bone changes and the initiation and progression of OA are still under debate. Previous studies have demonstrated that early (2-week OA) and late (12-week OA) application of extracorporeal shockwave therapy (ESWT) to the subchondral bone in the medial tibia condyle prior to development of OA changes exerted chondroprotective effects on the knee, with decreased cartilage degradation and improved subchondral bone remodeling, in a knee OA model in rats [3,4,5]. The results illustrated that ESWT can prevent or ameliorate the initiation or progressive change of OA in the knee when applied sooner. Many clinical trials have demonstrated that ESWT is safe and efficacious for the treatment of knee OA [6,7,8,9]; however, the mechanism of shockwave therapy on tissue regeneration in the human body remains unclear. Some studies have shown that ESWT promotes neovascularization (von Willebrand factor, vWF; vascular endothelial growth factor, VEGF; endothelial nitric oxide synthase, eNOS; and proliferating cell nuclear antigen, PCNA), bone-healing (bone morphogenetic protein-2, osteocalcin, alkaline phosphatase, dickkopf-related protein-1, and insulin-like growth factor), anti-inflammatory effects (Serum levels of soluble intercellular adhesion molecule and soluble vasccular cell adhesion molecule), and wound-healing (Wnt3, Wnt5a, and beta-catenin) [10,11,12,13,14]. ESWT also exhibits biological effects in terms of repairing bone by stimulation of expressions of eNOS, PCNA, VEGF, bone morphogenetic protein-2, and osteocalcin [3,11,15].

Mesenchymal stem cells (MSCs) can differentiate into various specialized cells, such as bone cells, cartilage, fat, cardiomyocytes, muscle fibers, and renal tubular cells, and may break germ layer commitment [16]. MSC treatment can be executed by direct replacement of injured tissue cells through the paracrine effect on the surrounding microenvironment, or indirectly by supporting revascularization, anti-apoptosis, and modulating the inflammatory response [17]. Recently, transplantation of ex vivo preparations of MSCs into the joints of animals with OA appeared to induce a therapeutically useful repair response as a result of paracrine responses of host cells, including progenitor populations residing within the synovium.

Bone marrow mesenchymal stem cells (BM-MSCs) have been well-studied and documented; however, they have been reported to have some limitations, such as low cell expansion, easy loss of stemness properties, and induction of artifactual chromosomal changes [18,19,20]. Adipose-derived mesenchymal stem cells (ADMSCs) from adipose tissue are a widely-used resource for therapy in regenerative medicine, and have been the focus of preclinical and clinical studies directed more towards numerous diseases than BM-MSCs [21,22]. The prodigious osteogenic potential of ADMSCs has been demonstrated in many preclinical animal studies [23]. Currently, many therapeutic regenerative strategies are being investigated as to whether autologous ADMSCs have significant effects on tissue regeneration and maintenance of articular cartilage in OA [24]. Combination therapy may offer extra benefits as compared with individual treatment in knee OA. During the experiments, we used two kinds of MSC (ADMSCs and Wharton’s jelly-derived mesenchymal stem cells (WJMSCs)) combined with ESWT to treat knee OA, and compared the effects of the treatments. In a previous study, we assessed the effects of combined ESWT and WJMSCs for the treatment of early knee OA [25]; however, different types of MSC have rarely been studied in terms of comparing their effectiveness in OA treatment. Therefore, in this study, we aimed to compare the effectiveness of ESWT and ADMSCs with that of ESWT and WJMSCs in an early rat knee OA model.

## 2. Results

### 2.1. Characterization of Human WJMSCs and Rat ADMSCs

It is important to characterize MSCs using cell-surface markers by flow cytometry analysis, but changes in cell markers need to be assessed with passaging. Passaging is considered to select the cell population with more homogenous cell-surface markers. Human WJMSCs express mesenchymal markers such as CD44, CD73, CD90, CD105, and CD166, and are negative for endothelial CD31 and hematopoietic CD14, CD45, CD34, and CD133. ADMSCs have been reported to express typical mesenchymal markers such as CD13, CD29, CD44, CD63, CD73, CD90, and CD105, and are negative for hematopoietic antigens such as CD14, CD31, CD45, and CD144 markers.

In the experiments, human WJMSCs and autologous rat ADMSCs were identified by specific cell-surface markers. WJMSCs showed a fibroblast-like morphology (Figure 1B). Flow cytometric analysis demonstrated that the WJMSCs positively expressed CD44 (98.65%), CD105 (88.86%), and CD166 (85.90%), and negatively expressed CD14 (18.93%) and CD133 (1.87%) (Figure 1C). A spindle-shaped, fibroblast-like morphology of the rat ADMSCs was observed, and showed typical findings, very similar to those of BM-MSCs (Figure 1D). The rat ADMSCs were found to express classic mesenchymal markers such as CD29 (100%) and CD90 (92.72%), as well as negative markers for hematopoietic antigens, including CD45, RT1a, RT1b, and CD106, by flow cytometric analysis (Figure 1E).

### 2.2. Injection of Human WJMSCs or Rat ADMSCs into OA Knee

Previous researchers have reported that MSCs can attach onto the articular cartilage via intra-articular injection and repair cartilage defects by differentiating into chondrocytes. Intra-articular injection of MSCs for OA therapy was found to be beneficial for animals and humans. Consequently, we focused the location of injection of WJMSCs or ADMSCs into the knee, and used ultrasound guidance to identify the location for injection of cells, which were injected from the upper position of the patella into the knee (Figure 2A). Imaging displayed the needle (red arrow) of the injector being placed into the knee of the rat smoothly by ultrasound machine guidance (Figure 2A); then, 1 × 10^6^ WJMSCs or ADMSCs were injected into the articular cavity of the knee.

### 2.3. ESWT Combined with Autologous Rat ADMSCs Synergistically Improved the Synovitis of Synovial Membrane and the Articular Cartilage in Early Knee OA

Implantation of stem cells in combination with ESWT has been reported as a new, alternative therapy for orthopedic diseases. In this study, we compared the results of xenografting of human WJMSCs and autologous rat ADMSCs combined with ESWT, and examined the synergistic effects of these treatments for early knee OA. After injection of MSCs, we applied ESWT to the medial tibia for the treatment of early knee OA (Figure 2B). An ultrasound machine was used to focus the location of the treatment, as described in previous studies. Each knee received 800 impulses of shockwave at an energy flux density of 0.25 mJ/mm^2^ in a single session (Figure 2B).

ESWT combined with human WJMSCs or autologous rat ADMSCs improved the damage to the articular cartilage for early treatment of osteoarthritis of the rat knee. In the WJMSC experiments, we used an anti-human-specific nuclei antigen antibody to detect human cells; however, no human cells were detected in the synovium or articular cartilage by immunohistochemistry (IHC) analysis post-treatment at 12 weeks, as was the case in our previous study. The synovitis of synovial membrane was significantly improved in OA+ESWT+ADMSCs compared with other groups (Figure 2C). Next, we observed changes to the articular cartilage caused by osteoarthritis by safranin O staining in the sham, OA, OA+ESWT, OA+WJMSCs, OA+ADMSCs, OA+ESWT+WJMSCs, and OA+ESWT+ADMSCs groups (Figure 2D). The OA+ESWT+ADMSCs group was the best improvement among the treatment groups in the synovitis scores. The protective effect on the articular cartilage was better in the OA+ESWT+ADMSCs group than in the other groups. The specific chondrocyte factors such as type II collagen and SOX-9 were measured to show the recovery of articular cartilage after treatments (Figure 2E,F). Notably, we found the expression of type II collagen in the OA+WJMSCs group to be much higher than the other groups, and the results were similar in our previous study. Our results demonstrated that ESWT combined with ADMSCs had better synergistic effects for protection of the synovial membrane and articular cartilage in the treatment of early knee OA in rats.

### 2.4. ESWT Combined with Autologous ADMSCs Was Better than Human WJMSCs in Improving Bone Volume of Early Knee OA

The results of micro-CT scanning are summarized in Figure 3. The subchondral plate thickness and bone volume were significantly decreased in the proximal tibia in the OA groups as compared with the sham group (*p* < 0.001). Treatment with ESWT and WJMSCs or ADMSCs significantly improved the subchondral plate thickness and bone volume in the OA+ESWT, OA+WJMSCs, and OA+ADMSCs groups. In addition, ESWT combined with WJMSCs or ADMSCs was more effective in terms of improvement of the subchondral plate thickness and bone volume than the other treatments (Figure 3B,C). We also observed that ESWT combined with ADMSCs was better than ESWT combined with WJMSCs with regards to improvement of bone volume (Figure 3C). Researchers reported previously that autologous MSCs can differentiate into cartilage and bone for the treatment of knee OA. The results demonstrated that the synergistic effects of ESWT combined with MSCs led to improvement in bone remodeling.

### 2.5. Effects of ESWT Combined with Autologous ADMSCs or Human WJMSCs to Regulate Specific Molecular Factors in the Treatment of Early Knee OA

Immunohistochemical analysis of TUNEL activity, caspase-3, transforming growth factor (TGF)-β, RUNX-2, PDGF-BB and BMP-4 in the articular cartilage of the sham, OA, OA+ESWT, OA+WJMSCs, OA+ESWT+WJMSCs, OA+ADMSCs, and OA+ESWT+ADMSCs groups were performed ( Figure 4 and Figure 5). The expressions of apoptosis markers, TUNEL activity, and caspase-3 were decreased in the OA+ESWT, OA+WJMSCs, OA+ESWT+WJMSCs, OA+ADMSCs, and OA+ESWT+ADMSCs groups as compared with the OA group (Figure 4). OA+ESWT+ADMSCs resulted in significantly improved levels of cartilage regeneration markers TGF-β, RUNX-2, and BMP-4 as compared with OA+ESWT, OA+WJMSCs, and OA+ESWT+WJMSCs, respectively (Figure 5). Further, PDGF-BB was significantly reduced with combined therapy (*p* < 0.001). These results demonstrated that ESWT combined with WJMSCs or ADMSCs synergistically modulated the key factors of joint repair in early OA of the knee. Finally, ESWT combined with autologous ADMSCs might be better than combined with WJMSCs in the treatment of early rat knee OA.

## 3. Discussion

In this study, we demonstrated that ESWT had a synergistic effect with autologous ADMSCs on early knee OA. We also found that ESWT combined with autologous ADMSCs was better than ESWT with xenografted WJMSCs in terms of joint repair in knee OA. The results showed that treatment of the rat knee OA with ESWT, autologous ADMSCs, WJMSCs, and combined therapy resulted in improvement in terms of histopathological examination, up-regulation of chondrogenic markers (RUNX-2, SOX-9, BMP-4), and down-regulation of apoptosis markers (TUNEL activity and caspase-3) in early knee OA in rats. Combined therapy displayed more effectiveness and efficacy with regards to cartilage and subchondral bone repair than ESWT and MSCs alone.

Hundreds of clinical trials have been registered to examine the effects of various MSCs; however, there remains a lack of consistent efficacy for the use of BM-MSCs, ADMSCs, and WJMSCs in the management of knee OA, and in particular, there have only been sporadic human trials performed using WJMSCs [25]. Cyclopedic preclinical studies have been performed, which have been aimed at unravelling the biological characteristics of MSCs in the treatment of OA. MSCs purified from different tissues have different paracrine effects depending on variable angiogenic, osteogenic, adipogenic, and chondrogenic potential properties [26,27]. Dr. Amable reported that WJMSCs secrete higher concentrations of chemokines, pro-inflammatory proteins, and growth factors, whereas ADMSCs exhibit a better pro-angiogenic profile and secrete higher amounts of extracellular matrix components and metalloproteinases in vitro [28]. In animal experiments, the efficacy and mechanism of MSCs in the treatment of OA remain elusive [29]. Prior studies indicated a more homogeneous, higher proliferative activity and chondrogenic potential of ADMSCs and WJMSCs than BM-MSCs in vivo [25]. Our data demonstrated no difference between WJMSC and ADMSC treatment in the synovitis scores, but WJMSCs had better histopathological OARSI scores following treatment than ADMSCs. This result might be correlated with high levels of type II collage in WJMSC treatment. However, it also may differ from human MSC and rat MSC in affecting the expression of key factors after treatment. Further, the expression levels of TUNEL activity, caspase-3, TGF-β, RUNX-2, SOX-9, PDGF-BB, or BMP-4 were different between the two groups of ADMSCs and WJMSCs (Figure 4 and Figure 5). However, in the combined therapy, ESWT with ADMSCs was better than with WJMSCs in knee OA treatment. This evidence shows that differences between the two cell therapies for OA may be influenced by other molecules or an unknown mechanism. Despite these efforts, the optimal preparation and mechanism of MSCs require further investigation.

The mechanisms of MSCs have been reported as contributing to tissue regeneration, including the paracrine effect, transdifferentiation, cell fusion, mitochondrial transfer, and exosome transfer [30]. Recent evidence indicated that MSCs serve as responder cells following transplantation, licensed by the host in order to secrete repair molecules. Moreover, after intra-articular delivery, low levels of MSC engraftment (typically 3% or less) accompanied by rapid clearance of the bulk population were observed [31,32,33]. In this study, xenografted human WJMSCs were delivered into rat knees with OA. After 12 weeks, there was no detectable anti-human-specific nuclei antigen antibody in the rat knees according to immunohistochemical analysis. This result may have been due to the xenografted human WJMSCs themselves having limited or no capability to directly restore, repair, or regenerate into tissue.

Combinations of distinct regeneration therapies have been vigorously studied in musculoskeletal disorders [34,35,36,37,38]. Dr. Saw treated grade 3 and 4 chondral lesions with arthroscopic subchondral drilling and postoperative intra-articular injection of hyaluronic acid. Additional peripheral blood stem cell injection resulted in better histologic and MRI evaluations [39]. Dr. Bastos reported that MSCs in combination with platelet-rich plasma (PRP) significantly improved the pain, function, daily living activities, and quality of life subscales in nine patients with symptomatic knee osteoarthritis [31], but the synergistic mechanism remains unclear. Some studies elaborated that oxygen tension, growth factor composition, and mechanical properties may serve to directly influence paracrine activity and regulate certain signaling pathways involved in chondrogenesis, chondral apoptosis, and osteogenesis [36,37,38,40,41]. ESWT is a kind of mechanical force, and the specific mechanism may be induced by stimulokinetics and stimulodynamics on tissue [42]. Dr. Zhang reported that shockwave promoted MSCs’ self-renewal and proliferation in repairing damaged cartilage [38]. Our prior animal experiment showed synergistic effects of ESWT and human WJMSCs in the improvement of bone volume and trabecular thickness, as well as in the expressions of RUNX-2, SOX-9, and collagen Xα1, which are important factors in chondrogenesis [25]. In this study, we identified more promising synergistic effects of using ESWT combined with autologous ADMSCs. The combination of ESWT and ADMSCs appeared to better improve bone volume, reinforce expressions of chondrogenic markers (TGF-β, RUNX-2, SOX-9, BMP-4), and down-regulate apoptosis markers (TUNEL activity and caspase-3) in early knee OA in rats (Figure 4). According to histological OARSI scores obtained via safranin O staining, however, ESWT combined with WJMSCs or ADMSCs had equal synergistic effects (Figure 4). This might have been due to the expression levels of cytokines and growth factors or other unknown factors. Additional experiments are needed to elucidate this phenomenon.

Many studies reported that WJMSCs are the candidate for the cellular therapies and allogenic transplantation because they are capable of immune suppression and immune avoidance. The reasons are as follows: (1) human leukocyte antigen (HLA) class I and an absence of HLA-DR are at a very low level of expression in WJMSCs [43,44]. (2) WJMSCs have been reported to modulate the suppression of T cells, which was in response to alloantigens [45]. (3) In the in vitro study, WJMSCs did not generate an immune response from allogeneic T cells [46]. All the studies revealed that WJMSCs had low immunogenicity and were suggested as being tolerated in allogeneic transplantation.

In our study, we also compared the effect of human WJMSCs and rat autologous ADMSCs on knee OA. Both MSCs had a chondroprotective effect on knee OA. However, we found the expression of type II collagen in the human WJMSCs group was higher than rat autologous ADMSCs group but not in the combined therapy (Figure 5A). The results are similar with our previous study and need to be further elucidated in the future [25]. Another question was proposed regarding if we were to use human ADMSCs in this experiment, could any difference between human ADMSC, rat ADMSCs, and human WJMSCs be shown? Dr. van Buul reported that there were no significant differences in the rat MSC, rat BM-MSCs, and human MSC on rat knee OA [47]. The results suggested that human MSC and rat MSC all have the chondroprotective effect, but the expression of the key factors may be difference on rat knee OA, such what is seen in the results in Figure 5.

This study had several limitations, as follows. First, it was a study that used small animals, and the results may differ in larger animal models or human clinical trials. Second, we assessed the results at the endpoint of 12 weeks; therefore, the duration of the therapeutic effect is still unclear. Third, the doses of shockwaves and MSCs were based on previous animal studies and references for cell-based therapies. These may not be the optimal doses or accurate cell preparations suitable for human clinical trials.

## 4. Materials and Methods

### 4.1. Experimental Animals

A total of 42 Sprague-Dawley rats (8 weeks old) were treated humanely according to the guidelines set down in the Guide for the Care and Use of Laboratory Animals, published by the National Institute of Health. All animals were housed under standard conditions. The Division of Laboratory Animal Resources at Chang Gung Memorial Hospital (CGMH), Kaohsiung Medical Center, provided veterinary care to the rodents. This study was subjected to the approval of the Institutional Animal Care and Use Committee (IACUC) on April 5, 2016 (Approval no. 2016011401). The study design and experiments are shown in Figure 1A.

### 4.2. Knee OA Was Induced by Anterior Cruciate Ligament Transection and Medial Meniscectomy

The left knees of the rats were prepared in a surgically sterile fashion. Through medial parapatellar mini-arthrotomy, the anterior cruciate ligament (ACL) was transected with a scalpel, and medial meniscectomy (MMx) was performed by removing the entire medial meniscus. The ACL was not transected in the sham group. The knee joint was irrigated and the incision closed, and prophylactic antibiotic treatment with ampicillin (Sigma-Aldrich, St. Louis, MO, USA) 50 mg/kg every 6 h was given for 5 days after surgery. Postoperatively, the animals were cared for by a veterinarian. The surgical site and the activities of the animals were observed daily.

### 4.3. ESWT Application

Animals undergoing ACLT+MMx in the ESWT group received ESWT 1 week after knee surgery in the experiments. The source of the focused shockwave was a DUOLITH SD1 (STORZ MEDICAL AG, Switzerland). The shockwave was focused on the subchondral bone of the medial tibia condyle of the left knee at 0.5 cm below the joint line and 0.5 cm from the medial skin surface. A surgical lubricate was used on the skin in contact with the shockwave device. Each knee was treated with 800 impulses of shockwave at an energy flux density of 0.25 mJ/mm^2^ in a single session [48]. After ESWT, the animals were returned to the housing cage for routine care and observation.

### 4.4. Rat Autologous ADMSCs and Human WJMSCs

Adipose tissue was minced and digested in 0.1% collagenase type I (GIBCO, Waltham, MA, USA) with shaking for 2 h at 37 °C. After digestion, an equal volume of Dulbecco’s modified Eagle’s medium (DMEM) containing 10% fetal bovine serum (FBS, Thermo Scientific, Waltham, MA, USA) was added and pipetted up and down several times. The cell suspension was filtered through a 100 μm filter (BD Falcon, San Jose, California, USA) to remove the solid aggregates. The sample was then centrifuged at 626× *g* for 5 min at 25 °C and vigorously mixed to complete separation of the stromal cells from the adipocytes. The centrifugation step was repeated and the pellet was re-suspended in 1 mL of lysis buffer (Promega, Walldorf, Baden-Wurttemberg, Germany) to lyse the red blood cells, and was then incubated for 10 min, washed with 10 mL of phosphate-buffered saline (PBS) with a 1% antibiotic-antimycotic mixture, and centrifuged at 626× *g* for 5 min. The supernatant was removed and the cell pellet was re-suspended in complete medium (DMEM with 20% FBS and 1% antibiotic-antimycotic solution) in a 25 cm^2^ culture flask and maintained in an incubator supplied with a humidified atmosphere of 5% CO_2_ at 37 °C.

WJMSCs were purchased from Cellular Engineering Technologies Inc. (item code: HMSC.WJ-100). The cells were cultured in DMEM-LG basal medium (Gibco, Waltham, MA, USA) and collected for subculture or stored for use after three to five passages. The ADMSCs or WJMSCs were injected into rat knee OA with 200 μL 1 × PBS containing 1 × 10^6^ cells (Figure 2A). After 30 min, ESWT was applied on the rat knee OA (Figure 2B).

### 4.5. Cell Phenotyping

The cellular morphology of the ADMSCs and WJMSCs became homogenously spindle-shaped in cultures after three to five passages. The specific surface molecules of the ADMSCs and WJMSCs were characterized by flow cytometry. Cells were detached with 0.05% Trypsin-EDTA in PBS and incubated with the respective antibody conjugated with fluorescein isothiocyanate or phycoerythrin against the following markers: ADMSCs with CD29, CD45, CD90, CD106, RT1a, and RT1b; WJMSCs with CD14, CD44, CD105, CD133, and CD166. Thereafter, the cells were analyzed using a flow cytometer (BD LSRII, Franklin Lakes, NJ, USA).

### 4.6. OARSI Score, Cartilage Area Measurement, and Synovitis Scoring

The degenerative changes of the cartilage were graded histologically using the Osteoarthritis Research Society International (OARSI) cartilage OA grading system via safranin O staining [49]. The scores were obtained on a scale of 0 to 24 by multiplying the index of the grade with the stage. For cartilage area measurement, eight non-consecutive sections, which were obtained at 100 μm intervals, were measured per femur condyle cartilage. Two reference points, 1 and 2, at a distance of 2.00 μm, which covered the majority of the cartilage layer, were automatically generated at the margin of the cartilage. The width of cartilage at a reference point was measured and the area was automatically calculated using imaging software. The synovial membrane was stained with hematoxylin and eosin to measure the synovitis scores. The features of chronic synovitis were described and defined as follows: (1) 0 to 1 = no synovitis; (2) 2 to 4 = lowgrade synovitis; (3) 5 to 9 = high-grade synovitis [25,50].

### 4.7. Micro-CT Scan

Collected specimens were subjected to micro-CT (SkyScan, 1076, Kartuizersweg 3B 2550. Kontich, Belgium) for bone analysis. The lower limb was prepared and sized to fit the micro-CT for scanning. The bone surface/volume ratio and trabecular thickness were measured, and were analyzed using a computer. Analyses of the trabecular bone included the trabecular volume fraction (BV/VT) and trabecular thickness.

### 4.8. Histopathological Examination

Cartilage and bone specimens were subjected to histopathological examination. The harvested specimens were fixed with 4% PBS-buffered formaldehyde at 4 °C for 7 days then decalcified in 10% PBS-buffered EDTA at 4 °C for 14 days. The decalcified specimens were fixed and subjected to paraffin wax embedding and dissection in 5-μm-thick sections. The specimens were stained with hematoxylin and eosin and safranin O stains. The degenerative changes of the cartilage were graded histologically using the Mankin score and OARSI score for assessment of cartilage structure, cartilage cells, and tidemark integrity.

### 4.9. Immunohistochemical Analysis

Knee specimens were further analyzed using immunohistochemical methods with anti-human specific nuclei antigen antibody, type II collagen, caspase-3, TGF-β, RUNX-2, SOX-9, PDGF-BB, and BMP-4. The specific inflammatory marker of TUNEL activity was also surveyed. The harvested specimens were fixed with 4% PBS-buffered formaldehyde for 48 h and decalcified in PBS-buffered 10% EDTA solution. The decalcified tissues were embedded in paraffin wax, and specimens were cut longitudinally in 5-μm-thick sections and transferred to poly-lysine-coated slides. Sections of the specimens were immunostained with specific reagents for anti-human-specific nuclei antigen antibody (Santa Cruz Biotechnology Inc., Dallas, Texas, USA) to identify the locations of the WJMSCs. Anti-rat caspase-3, type II collagen, TGF-β, RUNX-2, SOX-9, PDGF-BB, and BMP-4 (Abcam Company, Eugene, OR, USA) are markers of cartilage repair of rats. The immunoreactivity of specimens was demonstrated using a horseradish peroxidase (HRP)-3′-, 3′-diaminobenzidine (DAB) cell and tissue staining kit (R&D Systems, Inc., Minneapolis, MN, USA). The immunoreactivities were quantified from five areas in three sections of the same specimen using a Zeiss Axioskop 2 plus microscope (Carl Zeiss, Oberkochen, Germany). All images of each specimen were captured using a Cool CCD camera (SNAP-Pro c.f. Digital kit; Media Cybernetics, Silver Spring, MD, USA). Images were analyzed using Image-Pro Plus image analysis software (Media Cybernetics, Rockville, Maryland, USA). The percentages of immuno-labeled positive cells over the total cells in each area were calculated and the average of each specimen was used as the result.

### 4.10. Statistical Analysis

Data obtained before and after treatment within the same group were compared statistically using a paired *t*-test. Data of the study group and the control group were compared statistically using the Mann–Whitney U test. Data of different groups were compared statistically using the chi-squared test. Statistical significance was set at *p* < 0.05, 0.01, and 0.001.

## 5. Conclusions

Combined ESWT with autologous ADMSCs was found to be effective for the treatment of early knee OA in rats. Further, we demonstrated that ESWT with autologous ADMSCs had synergistic effects greater than those of ESWT, MSCs, and ESWT with WJMSCs in the treatment of early knee OA. These findings provided insights for innovative strategies using combined ESWT with ADMSCs for the treatment of knee OA.

## Figures and Tables

**Figure 1 ijms-21-01217-f001:**
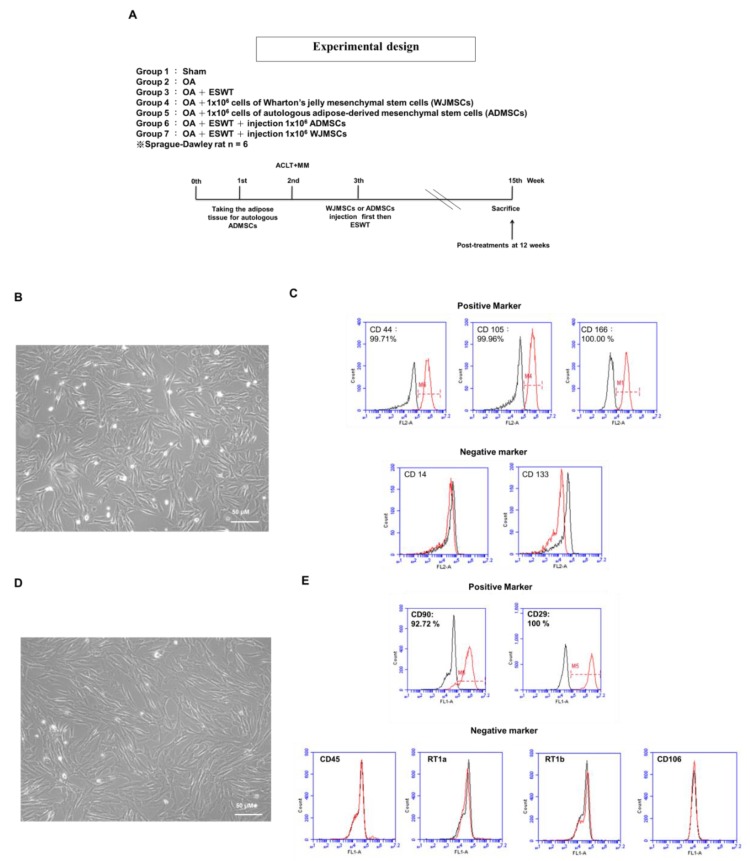
Study design and characterization of human WJMSCs and rat ADMSCs. (**A**) Graphic scheme depicting the study design of the experiment, including knee surgery, shockwave application, injection of Wharton’s jelly-derived mesenchymal stem cells (WJMSCs) or adipose-derived mesenchymal stem cells (ADMSCs), and sacrifice of animals. Six rats in each group were used in the experiments. (**B**) Morphology of cultured WJMSCs. (**C**) Analysis of cell-surface markers on WJMSCs. Cells were labeled with the indicated markers: CD44 (99.71%), CD105 (99.96%), and CD166 (100.00%) as positive markers; CD14 and CD133 as negative markers. (**D**) Morphology of cultured rat ADMSCs. (**E**) Analysis of cell-surface markers on rat ADMSCs. Cells were labeled with the indicated markers: CD90 (92.72%) and CD29 (100.00%) as positive markers; CD45, RT1a, RT1b, and CD106 as negative markers. The percentage is shown as the red line; the black line represents isotype controls.

**Figure 2 ijms-21-01217-f002:**
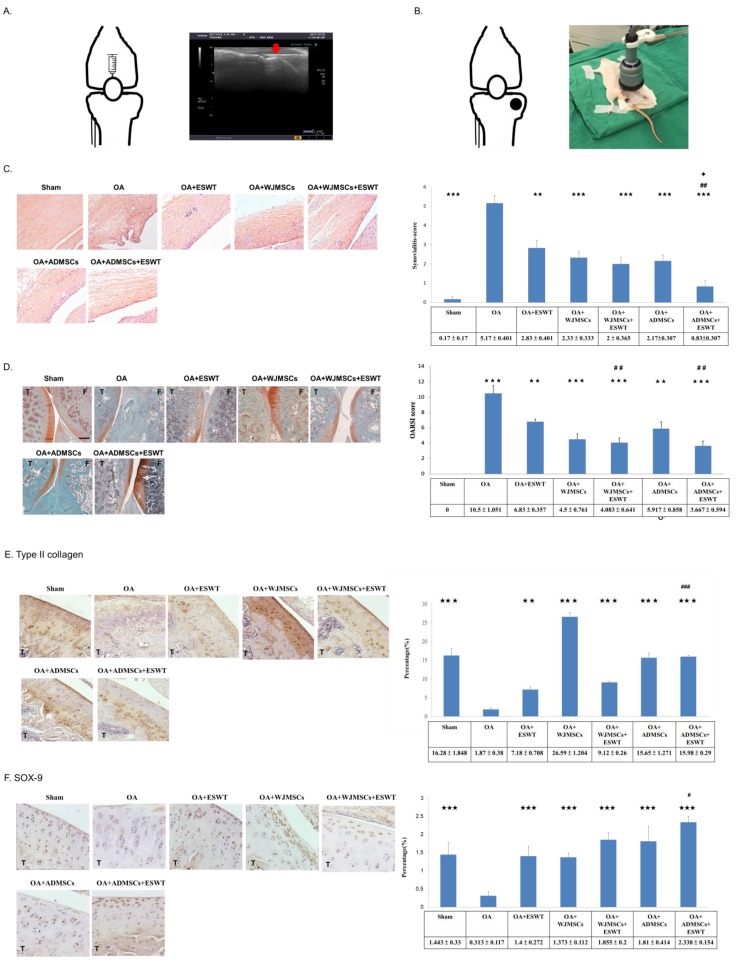
Treatments, histological analysis of synovium membrane, and Osteoarthritis Research Society International (OARSI) scores of the knee osteoarthritis (OA). (**A**) Sketch of the knee outlining the location of injection of WJMSCs or ADMSCs in the rats. In the left, ultrasound guidance (Toshiba Medical Systems Corporation, Tokyo, Japan) was used for localization, and WJMSCs or ADMSCs were injected into the knees of the rats. The red arrow indicates the needle of the injector. (**B**) Knee sketch outlining the locations (black dots) of shockwave application in the different groups of animals. The left picture showed that extracorporeal shockwave therapy (ESWT) was applied to the knees of the rats; *n* = 6. (**C**) The synovium membrane was displayed by hematoxylin and eosin staining at magnifications of 200×. The left panel shows synovitis scores were measured in each group. (**D**) Microphotographs of cartilage and subchondral bone showed changes in cartilage damage in the OA group and protection in the treatment groups by safranin O staining. (**E**) Type II collagen (×200 magnification). (**F**) SOX-9 (×400 magnification). Scale bar represents 200 μm; T indicates the tibia; F indicates the femur. ** *p* < 0.01, and *** *p* < 0.001 were compared with OA; ^#^
*p* < 0.05, ^##^
*p* < 0.01, and ^###^
*p* < 0.001 were compared among treatment groups. ✦ *p* < 0.05 was compared ESWT combined with ADMSCs or WJMSCs each other. *n* = 6 for each group.

**Figure 3 ijms-21-01217-f003:**
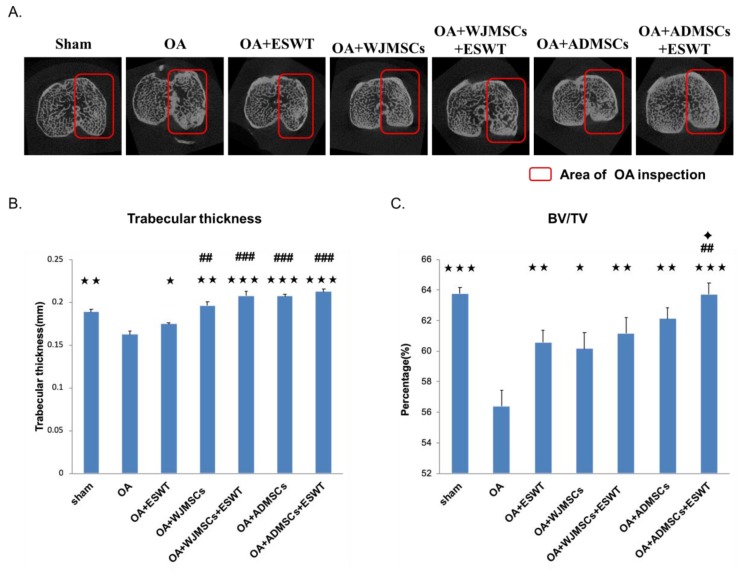
Micro-CT scans of the proximal tibia in different groups. (**A**) Photomicrographs of the knee in the sagittal and transverse views by micro-CT. The subchondral bone medial compartment of each group is marked by a red box. (**B**) Data resulting in graphic illustrations of trabecular thickness and (**C**) percentage of trabecular bone volume fraction (BV/TV) in the different groups. * *p* < 0.05, ** *p* < 0.01, and *** *p* < 0.001 were compared with OA; ^##^
*p* < 0.01, and ^###^
*p* < 0.001 were compared among treatment groups. ✦ *p* < 0.05 was compared ESWT combined with ADMSCs or WJMSCs each other. *n* = 6 for all groups.

**Figure 4 ijms-21-01217-f004:**
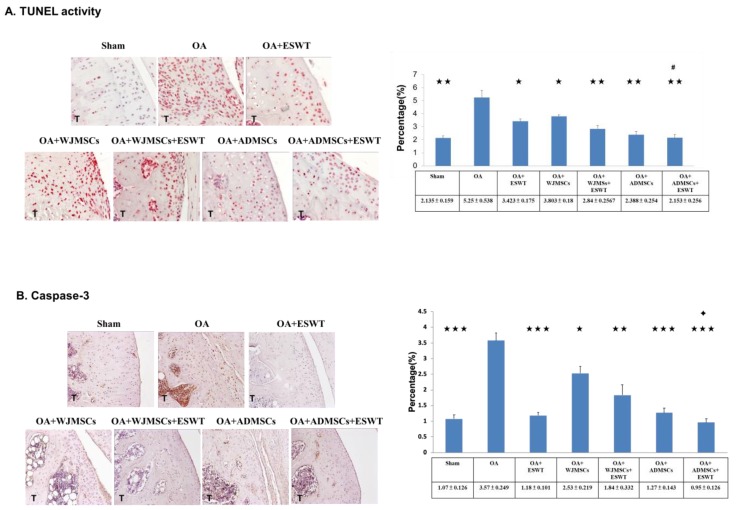
Immunohistochemical analysis for apoptosis markers. (**A**) TUNEL activity (×400 magnification), (**B**) caspase-3 in the experiments (left) (×200 magnification), and the level of expression that was measured after treatment (right). * *p* < 0.05, ** *p* < 0.01, and *** *p* < 0.001 were compared with OA; ^#^
*p* < 0.05 was compared among treatment groups. ✦ *p* < 0.05 was compared between ESWT combined with ADMSCs or WJMSCs. All rats were *n* = 6. Tibia is indicated as T.

**Figure 5 ijms-21-01217-f005:**
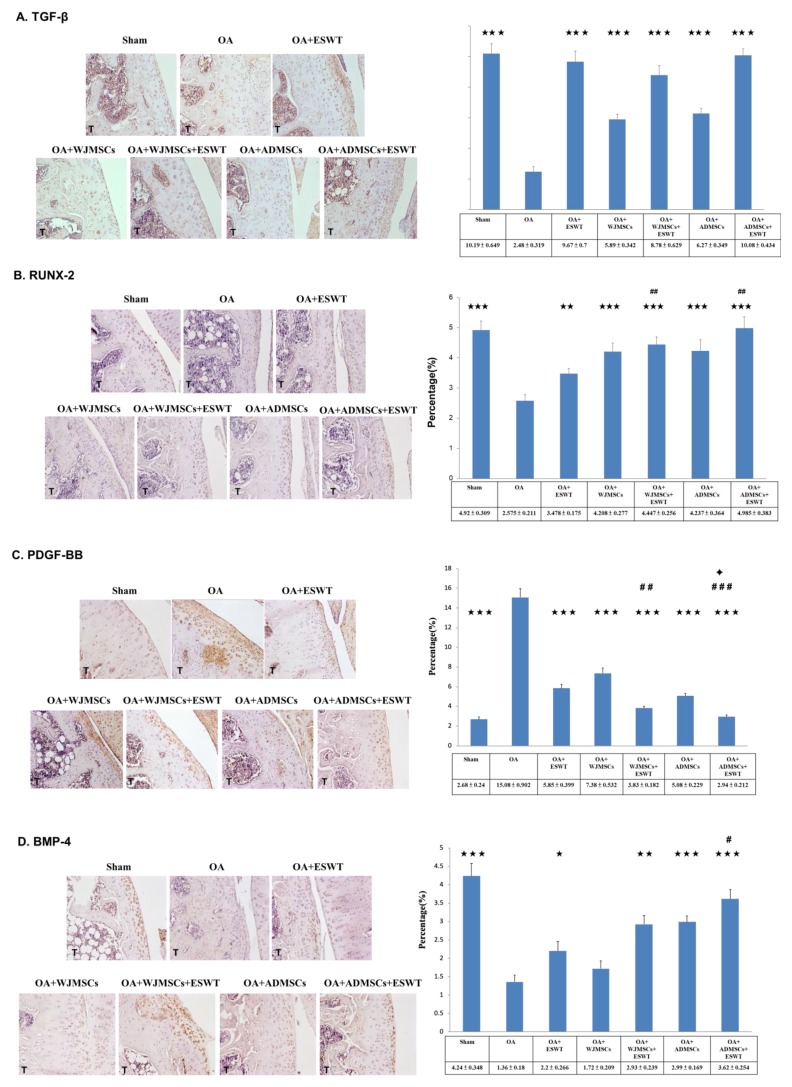
Immunohistochemical analysis for cartilage development-specific markers. (**A**) TGF-β (×200 magnification), (**B**) RUNX-2 (×200 magnification), (**C**) PDGF-BB (×200 magnification), and (**D**) BMP-4 in the experiments (×200 magnification) (left), and levels of expression measured after treatment (right). * *p* < 0.05, ** *p* < 0.01, and *** *p* < 0.001 were compared with OA; ^#^
*p* < 0.05, ^##^
*p* < 0.01, and ^###^
*p* < 0.001 were compared among treatment groups. ✦ *p* < 0.05 was compared between ESWT and combined ADMSCs or WJMSCs. *n* = 6 for all groups. T indicates the tibia.

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
