# Peer review of "Shockwave Therapy Combined with Autologous Adipose-Derived Mesenchymal Stem Cells Is Better than with Human Umbilical Cord Wharton’s Jelly-Derived Mesenchymal Stem Cells on Knee Osteoarthritis"

_ijms, 2020, doi:10.3390/ijms21041217_

Round 1

Reviewer 1 Report

The authors compared the effect of adipose-derived mesenchymal stem cells (AD-MSCs) with those of Wharton’s jelly-derived mesenchymal stem cells (WJMSCs) in knee osteoarthritis (OA) treated with extracorporeal shockwave therapy (ESWT) in early rat OA model.

Major comments:

The authors used human WJMSCs and rat AD-MSCs. If the authors used human AD-MSCs in this rat OA model, could it be shown any difference between human AD-MSC and rat AD-MSCs? In other words, human AD-MSC and rat AD-MSCs are comparable in chondroprotective effect? Is there any difference between human WJMSCs and rat AD-MSCs in terms of chondrogenic potential or immune regulatory effect? (Please add some comments in Discussion section.)

Author Response

Manuscript Submissions

Editor of International Journal of Molecular Sciences

Dear Editor,

The title of the manuscript was modified by the sugesstion of reviewer 1 as “The Shockwave Therapy Combined with Autologous Adipose-Derived Mesenchymal Stem Cells are better than with Human Umbilical Cord Wharton’s Jelly-Derived Mesenchymal Stem Cells on Knee Osteoarthritis”. In this manuscript, all authors agree to revise and submit the revision to International Journal of Molecular Sciences. The authors appreciated the suggestions of editor and reviewers. We were point-by-point to response the comments as below.

Reviewer 1:

Comments and Suggestions for Authors

The authors compared the effect of adipose-derived mesenchymal stem cells (AD-MSCs) with those of Wharton’s jelly-derived mesenchymal stem cells (WJMSCs) in knee osteoarthritis (OA) treated with extracorporeal shockwave therapy (ESWT) in early rat OA model.

Major comments:

The authors used human WJMSCs and rat AD-MSCs. If the authors used human AD-MSCs in this rat OA model, could it be shown any difference between human AD-MSC and rat AD-MSCs? In other words, human AD-MSC and rat AD-MSCs are comparable in chondroprotective effect? Is there any difference between human WJMSCs and rat AD-MSCs in terms of chondrogenic potential or immune regulatory effect? (Please add some comments in Discussion section.)

Response: Thank revierer’s suggestion. We discussed the issue in the Discussion section. Further, the immunohistochemical staining was displayed as below. There were no human-specific nuclei markers in the rat knee post-treatment at 12 weeks (Articular cartilage, subchondral bone and synovial membrane) and were displayed in the human skin specimen (Arrow). The similar result for measuring the human stem cell in rat was reported by using human ADMSCs in the treatment of rat knee OA [1].

Li, M.; Luo, X.; Lv, X.; Liu, V.; Zhao, G.; Zhang, X.; Cao, W.; Wang, R.; Wang, W., In vivo human adipose-derived mesenchymal stem cell tracking after intra-articular delivery in a rat osteoarthritis model. Stem Cell Research & Therapy 2016, 7, (1).

Reviewer 2 Report

It could be reconsider after major revision, as the controls are missing and figures need a substantial improvement.

1) The title is very long and confusing. It could be shorter and more informative.

2) figure 1 and 2 should be merged.

3) in the graphic scheme of figure 1, the word week should be written on the side, as it indicates the number of weeks and does not just belong to the 0th week.

4) in figure 2, the immune-histopathological images should be added. By the way, for both WJ and AD-MSCs, the classical way of showing the intrinsic potential of MSC differentiation to adipocytes, chondrocytes and osteoblasts should be shown.

5) Figures 3 and 4 should be merged as they are interdependent and Figure 3 alone cannot justify presentation as an independent figure.

6) Figure 4: only panel A is described in the figure legend. Furthermore, histopathological images are of very low quality, no magnification has been shown. This illustration needs a substantial improvement.

6) Figure 4: The stained sections should be from the same region and in a sequential series so that the reader can see which region has been stained for H&E and what the positivity or negativity of this specific region is for the saffranin-o staining. In addition, these sections should be stained for at least two additional chondrocyte factors to make chondrogenesis conform. The images should be shown at 100x and 200x magnification.

7) Figures 6 and 7 are a big confusion. The images should be shown in a row and should be from the same region to facilitate a statement. In this way it is very difficult to make a statement from the pictures. It is very hard to accept that pictures are taken at 400x. In the current presentation, it is very difficult to make a statement. pictures need a better quality and magnification.

8) The positive and negative controls are missing. Positive could be bone marrow MSC and negative any other cell type like epithelial cells.

Author Response

Manuscript Submissions

Editor of International Journal of Molecular Sciences

Dear Editor,

The title of the manuscript was modified by the sugesstion of reviewer 1 as “The Shockwave Therapy Combined with Autologous Adipose-Derived Mesenchymal Stem Cells are better than with Human Umbilical Cord Wharton’s Jelly-Derived Mesenchymal Stem Cells on Knee Osteoarthritis”. In this manuscript, all authors agree to revise and submit the revision to International Journal of Molecular Sciences. The authors appreciated the suggestions of editor and reviewers. We were point-by-point to response the comments as below.

Reviewer 2:

Comments and Suggestions for Authors

It could be reconsider after major revision, as the controls are missing and figures need a substantial improvement.

1) The title is very long and confusing. It could be shorter and more informative.

Response: Thank suggestion. We modified and shorten the title in the manuscript.

2) figure 1 and 2 should be merged.

Response: Thank suggestion. We merged the old Figure 1 and Figure 2 into new Figure 1.

3) in the graphic scheme of figure 1, the word week should be written on the side, as it indicates the number of weeks and does not just belong to the 0th week.

Response: We verified the number of week in the new Figure 1A.

4) in figure 2, the immune-histopathological images should be added. By the way, for both WJ and AD-MSCs, the classical way of showing the intrinsic potential of MSC differentiation to adipocytes, chondrocytes and osteoblasts should be shown.

Response: Thank suggestion. We measured the surface markers of stem cells to make sure the characteristics of MSC. The human WJMSCs were measured post-treatment at 12 weeks. The immunohistochemical staining was displayed as bellow. There were no human-specific nuclei markers in the rat knee post-treatment at 12 weeks (Articular cartilage, subchondral bone and synovial membrane) and were displayed in the human skin specimen (Arrow). The similar result for measuring the human stem cell in rat was reported by using human ADMSCs in the treatment of rat knee OA[1].

Li, M.; Luo, X.; Lv, X.; Liu, V.; Zhao, G.; Zhang, X.; Cao, W.; Wang, R.; Wang, W., In vivo human adipose-derived mesenchymal stem cell tracking after intra-articular delivery in a rat osteoarthritis model. Stem Cell Research & Therapy 2016, 7, (1).

5) Figures 3 and 4 should be merged as they are interdependent and Figure 3 alone cannot justify presentation as an independent figure.

Response: Thanks. We merged the old figure 3 and figure 4 into new Figure 2 in the manuscript.

6) Figure 4: only panel A is described in the figure legend. Furthermore, histopathological images are of very low quality, no magnification has been shown. This illustration needs a substantial improvement.

Response: We corrected the mistakes and improved the quality of the figure.

6) Figure 4: The stained sections should be from the same region and in a sequential series so that the reader can see which region has been stained for H&E and what the positivity or negativity of this specific region is for the saffranin-o staining. In addition, these sections should be stained for at least two additional chondrocyte factors to make chondrogenesis conform. The images should be shown at 100x and 200x magnification.

Response: Thank suggestion. Now, old figure 4 was merged into the new figure 2 C and the regions were synovium membrane of the knee. The synovitis scores were measured after treatments on knee OA. The factors of chondrogenesis were observed in the figure 5. The typos were verified in the figure legend.

7) Figures 6 and 7 are a big confusion. The images should be shown in a row and should be from the same region to facilitate a statement. In this way it is very difficult to make a statement from the pictures. It is very hard to accept that pictures are taken at 400x. In the current presentation, it is very difficult to make a statement. pictures need a better quality and magnification.

Response: We rearranged the Figures and improve the quality of the figures. The mistakes were verified carefully.

8) The positive and negative controls are missing. Positive could be bone marrow MSC and negative any other cell type like epithelial cells.

Response: It’s a grateful for the reviewer’s suggestion. In the experiments, we measured the treatments by compared with OA (damage) and Sham (normal) groups to observe the chondroprotective effect. We will add the bone marrow MSC and epithelial cells as controls in the next experiment. 

Round 2

Reviewer 1 Report

The authors revised the manuscript as it desired. 

Author Response

The title of the manuscript was modified by the sugesstion of reviewers as “The Shockwave Therapy Combined with Autologous Adipose-Derived Mesenchymal Stem Cells are better than with Human Umbilical Cord Wharton’s Jelly-Derived Mesenchymal Stem Cells on Knee Osteoarthritis”. In this manuscript, all authors agree to revise and submit the revision to International Journal of Molecular Sciences. The authors appreciated the suggestions of editor and reviewers. We were point-by-point to response the comments as below. In this major revision, we would like to thank the reviewer1 for careful and thorough reading of this manuscript and for the thoughtful comments and constructive suggestions.

Reviewer 1:

Comments and Suggestions for Authors

The authors compared the effect of adipose-derived mesenchymal stem cells (AD-MSCs) with those of Wharton’s jelly-derived mesenchymal stem cells (WJMSCs) in knee osteoarthritis (OA) treated with extracorporeal shockwave therapy (ESWT) in early rat OA model.

Major comments:

The authors used human WJMSCs and rat AD-MSCs. If the authors used human AD-MSCs in this rat OA model, could it be shown any difference between human AD-MSC and rat AD-MSCs? In other words, human AD-MSC and rat AD-MSCs are comparable in chondroprotective effect? Is there any difference between human WJMSCs and rat AD-MSCs in terms of chondrogenic potential or immune regulatory effect? (Please add some comments in Discussion section.)

Response: Thank revierer’s suggestion. We discussed the issue in the Discussion section. Further, the immunohistochemical staining was displayed as below. There were no human-specific nuclei markers in the rat knee post-treatment at 12 weeks (Articular cartilage, subchondral bone and synovial membrane) and were displayed in the human skin specimen (Arrow). The similar result for measuring the human stem cell in rat was reported by using human ADMSCs in the treatment of rat knee OA [1].

Li, M.; Luo, X.; Lv, X.; Liu, V.; Zhao, G.; Zhang, X.; Cao, W.; Wang, R.; Wang, W., In vivo human adipose-derived mesenchymal stem cell tracking after intra-articular delivery in a rat osteoarthritis model. Stem Cell Research & Therapy 2016, 7, (1).

Reviewer 2 Report

Surprisingly, the authors ignored almost all the proposed comments and forgot to answer the questions. I would say that in this format and with the remaining questions being fundamental, this paper cannot be granted for publication.

Author Response

Dear Reviewer2,

The title of the manuscript was modified by the sugesstion as “The Shockwave Therapy Combined with Autologous Adipose-Derived Mesenchymal Stem Cells are better than with Human Umbilical Cord Wharton’s Jelly-Derived Mesenchymal Stem Cells on Knee Osteoarthritis”. In this manuscript, all authors agree to revise and submit the revision to International Journal of Molecular Sciences. The authors appreciated the suggestions of reviewer2. We were point-by-point to response the comments as below.

Comments and Suggestions for Authors

It could be reconsider after major revision, as the controls are missing and figures need a substantial improvement.

1) The title is very long and confusing. It could be shorter and more informative.

Response: We would like to thank the reviewer2 for the suggestion. We modified and shorten the title in the manuscript.

2) figure 1 and 2 should be merged.

Response: We appreciate for the reviewer's recommendation. The original Figure 1 and Figure 2 have been edited and merged into new Figure 1.

3) in the graphic scheme of figure 1, the word week should be written on the side, as it indicates the number of weeks and does not just belong to the 0th week.

Response: Thanks for the reviewer’s reminding. The revisions and suggestions have been added in the new Figure 1A.

4) in figure 2, the immune-histopathological images should be added. By the way, for both WJ and AD-MSCs, the classical way of showing the intrinsic potential of MSC differentiation to adipocytes, chondrocytes and osteoblasts should be shown.

Response: Thanks for the reviewer’s reminding, to be honest, we haven’t used the differentiation medium to test the intrinsic potential of MSC. We will do these experiments for a couple weeks, but not really for the 10 days revision. We are sorry for our imperfect experiment. However, we measured the composition of CD markers typical for MSC by flow cytometry analysis.

In another way, the human WJMSCs were measured post-treatment at 12 weeks. The immunohistochemical staining was displayed as below. There were no human-specific nuclei markers in the rat knee post-treatment at 12 weeks (Articular cartilage, subchondral bone and synovial membrane) but were displayed in the human skin specimen (Arrow). The similar result for measuring the human stem cell in rat was reported by using human ADMSCs in the treatment of rat knee OA[1]. The results indicated that MSCs were not differentiation or did not exist for a long time in acceptor tissue.

Li, M.; Luo, X.; Lv, X.; Liu, V.; Zhao, G.; Zhang, X.; Cao, W.; Wang, R.; Wang, W., In vivo human adipose-derived mesenchymal stem cell tracking after intra-articular delivery in a rat osteoarthritis model. Stem Cell Research & Therapy 2016, 7, (1).

5) Figures 3 and 4 should be merged as they are interdependent and Figure 3 alone cannot justify presentation as an independent figure.

Response: We thank the reviewer for this valuable suggestion. We merged the original figure 3 and figure 4 into new Figure 2 in the manuscript.

6) Figure 4: only panel A is described in the figure legend. Furthermore, histopathological images are of very low quality, no magnification has been shown. This illustration needs a substantial improvement.

Response: Thank you very much for reviewer suggestion. We corrected the mistakes and improved the quality of the figure.

6) Figure 4: The stained sections should be from the same region and in a sequential series so that the reader can see which region has been stained for H&E and what the positivity or negativity of this specific region is for the saffranin-o staining. In addition, these sections should be stained for at least two additional chondrocyte factors to make chondrogenesis conform. The images should be shown at 100x and 200x magnification.

Response: Many thanks for the reviewer. Now, original figure 4 was merged into the new figure 2 C and the regions were synovium membrane of the knee. We focused the level of inflammation in the synovium membrane after treatment by HE stain. The synovitis scores were measured after treatments for knee OA. The damage regions of articular cartilage were stained by Saffranin-O and measured the level of damage by OARSI score in the figure 2D. We added the factors of chondrogenesis in the figure 2E and 2F such as type II collagen, SOX9 to make chondrogenesis conform. The typos were verified in the figure legend.

7) Figures 6 and 7 are a big confusion. The images should be shown in a row and should be from the same region to facilitate a statement. In this way it is very difficult to make a statement from the pictures. It is very hard to accept that pictures are taken at 400x. In the current presentation, it is very difficult to make a statement. pictures need a better quality and magnification.

Response: Thank you very much for your beneficial comments. The original figure 6 was observed the expression of apoptosis markers including TUNEL activity and Caspase 3. Now, we rearranged the Figure 6 into Figure 4 and improved the quality of the figure. We displayed the levels of apoptosis factors and regeneration factors in the articular cartilage after treatments separately in the Figure 4 and Figure 5. The mistakes were verified carefully.

8) The positive and negative controls are missing. Positive could be bone marrow MSC and negative any other cell type like epithelial cells.

Response: It’s a grateful for the reviewer’s suggestion. In the experiments, we measured the treatments by compared with OA (damage) and Sham (normal) groups to observe the chondroprotective effect as our previous studies. The bone marrow MSC was not the study target of this project and we did not have the bone marrow MSC in our laboratory. We will consider the bone marrow MSC and epithelial cells as controls in the next project. We are grateful to the reviewer2 for the insightful comments on this paper.
